# Pathomechanisms of Non-Traumatic Acute Brain Injury in Critically Ill Patients

**DOI:** 10.3390/medicina56090469

**Published:** 2020-09-13

**Authors:** Wojciech Dabrowski, Dorota Siwicka-Gieroba, Malgorzata Gasinska-Blotniak, Sami Zaid, Maja Jezierska, Cezary Pakulski, Shawniqua Williams Roberson, Eugene Wesley Ely, Katarzyna Kotfis

**Affiliations:** 1Department of Anaesthesiology and Intensive Care, Medical University of Lublin, 20-954 Lublin, Poland; dsiw@wp.pl (D.S.-G.); gasinskamalgorzata@wp.pl (M.G.-B.); majajezierska1991@gmail.com (M.J.); 2Department of Anaesthesia, Al-Emadi-Hospital Doha, P.O. Box 5804 Doha, Qatar; sami1zaid@gmail.com; 3Department of Anaesthesiology, Intensive Therapy and Emergency Medicine, Pomeranian Medical University in Szczecin, 71-252 Szczecin, Poland; cezary.pakulski1@gmail.com; 4Critical Illness, Brain Dysfunction, and Survivorship (CIBS) Center, Vanderbilt University Medical Center, 1211, Nashville, TN 37232, USA; shawniqua.w.roberson@vumc.org (S.W.R.); wes.ely@vumc.org (E.W.E.); 5Department of Neurology, Vanderbilt University Medical Center, 1211, Nashville, TN 37232, USA; 6Department of Biomedical Engineering, Vanderbilt University, 1211, Nashville, TN 37232, USA; 7Geriatric Research, Education and Clinical Center (GRECC), Tennessee Valley Veterans Affairs Healthcare System, 1310, Nashville, TN 37212, USA; 8Division of Allergy, Pulmonary, and Critical Care Medicine, Vanderbilt University Medical Center, 1211, Nashville, TN 37232, USA; 9Department of Anaesthesiology, Intensive Therapy and Acute Intoxications, Pomeranian Medical University, 70-111 Szczecin, Poland

**Keywords:** delirium, critical illness, hypoxia, hyperoxia, neuroinflammation, neurotransmitter agents, kynurenine pathway, tryptophan, gastrointestinal microbiome

## Abstract

Delirium, an acute alteration in mental status characterized by confusion, inattention and a fluctuating level of arousal, is a common problem in critically ill patients. Delirium prolongs hospital stay and is associated with higher mortality. The pathophysiology of delirium has not been fully elucidated. Neuroinflammation and neurotransmitter imbalance seem to be the most important factors for delirium development. In this review, we present the most important pathomechanisms of delirium in critically ill patients, such as neuroinflammation, neurotransmitter imbalance, hypoxia and hyperoxia, tryptophan pathway disorders, and gut microbiota imbalance. A thorough understanding of delirium pathomechanisms is essential for effective prevention and treatment of this underestimated pathology in critically ill patients.

## 1. Introduction

Acute non-traumatic brain injury, manifested as a type of neuropsychological dysfunction, is frequently noted in critically ill patients undergoing elective or emergency surgery and treated in the intensive care unit (ICU). It includes various types of behavioral disorders, commonly known as delirium, defined as an acute disturbance in attention and awareness with additional disturbances in cognition not resulting from pre-existing neuropsychological disorders, and caused by another medical condition [1,2]. Delirium has been observed in patients undergoing elective or emergency surgery, and in critically ill patients treated in the ICU. The rates of delirium vary between 68% and 80% in mechanically ventilated critically ill patients and between 9% and 70% in patients without artificial breathing support [3,4,5]. It has been documented that delirium contributes independently to poor outcome and prolongs the length of stay in the hospital [6,7]. The fact that delirium is a strong determinant of hospital stay and is an extremely common complication of ICU treatment implicates it as a contributing factor to the increased cost of hospitalization.

Generally, delirium is classified into three main subtypes: hyperactive and its extreme subtype, excitation, hypoactive and its extreme subtype, a form of catatonia, and mixed [8]. Based on clinical manifestations, delirium has five core domains: psychomotor disturbance, emotional dysregulation, cognitive deficits, attention deficits and disorders in circadian rhythm [9]. Importantly, hypoactive delirium is associated with worse long-term cognition than the hyperactive subtype [7]. Many patients remain undiagnosed when validated delirium screening tools are not used, both in the ICU and outside the ICU. Therefore, delirium monitoring policies should be implemented in every hospital ward [10].

More than 100 different risk factors have been described for delirium, and are categorized into predisposing factors (present before the admission to the hospital) and potentially modifiable precipitating factors (generated during the treatment period) [11]. Despite a large number of risk factors, only a few of them significantly increase the incidence of delirium in hospital settings. Hypoxia, prolonged mechanical ventilation, hyperbilirubinemia, raised creatinine and the use of benzodiazepines, as well as elderly age, sleep deprivation, alcohol or drug addiction, physical immobility, severe comorbidities and severe infection predispose to the development of delirium [8,9,12].

Pathomechanisms of acute non-traumatic brain injury have been studied for many recent years. Currently, there are a few leading theories that seem to explain the non-traumatic brain damage. Some of them seem to be very clear, whereas others are very controversial (Table 1). The aim of this review was to analyze the most popularly known pathomechanisms of delirium in critically ill patients treated in the ICU.

## 2. Hypoxia or Hyperoxia-Related Brain Injury

The brain is considered the most vulnerable organ at the highest risk of oxygen disorders. When oxygen delivery to the brain is decreased below a critical value, a biochemical cascade is induced that leads to neuronal damage. Cerebral hypoxia results in changes in the intra- and extracellular electrolyte concentrations. An anoxia-related increase in cell membrane permeability occurs between 60 and 180 s after the onset, leading to a decrease in extracellular sodium, chloride and calcium with an increase in potassium leaks from the neuronal cells. At the same time, calcium rapidly influxes into the neurons, leading to subsequent mitochondrial dysfunction and overproduction of reactive oxygen radicals [13]. The mitochondrial dysfunction causes further ATP depletion, which impairs osmotic pump activity. These disorders may induce neuronal apoptosis and necrosis within a few hours; necrotic neuronal damage is commonly observed early after severe ischemic events, whereas apoptotic cell death may occur with longer survival periods [14]. Additionally, prolonged or intermittent hypoxia activates microglia, which is a trigger for neuroinflammation manifested as a so-called delayed post-anoxic encephalopathy [15]. A decrease in cerebral oxygen saturation is associated with delirium in septic shock patients [16]. Mikkelsen and colleagues also found relationships between lower PaO_2_ on the day of admission to the ICU and cognitive impairment in general, and executive dysfunction specifically [17]. The duration of hypoxemia during admission correlated with attention, verbal memory and executive function in patients treated for severe acute respiratory distress syndrome with PaO_2_/FiO_2_ < 150 mmHg, but it did not correlate with any neurocognitive functions at two-year follow-up [18].

Similar to hypoxia, hyperoxia may also be harmful and increases the risk for acute non-traumatic brain injury [19,20]. The most dramatic disorders in brain function and cerebral blood flow were observed in healthy volunteers with combined hyperoxia and hypocapnia during anesthesia [20]. Notably, hyperoxia induces hypocapnia following hyperventilation, which is explained by the Haldane effect (oxygenated hemoglobin binds less CO_2_). Unbounded CO_2_ must be transported as the dissolved ion, which increases pH and stimulates the brainstem nuclei to increase ventilation. Exposure to arterial hyperoxia correlates with worse outcome and higher mortality in stroke and traumatic brain injury patients [21]. Hyperoxia decreases phosphorylation of protein 3-kinase and increases activation of c-Jun N-terminal kinase, favoring apoptotic cell death [22]. Acute 6-hour hyper-oxygenation markedly down-regulates the brain-derived neurotrophic factors, neutrophins 3 and 4, and induces oxidative stress, leading to apoptotic neurodegeneration [23]. High blood oxygen tension increases F_2_-isoprostane and isofuran plasma concentrations—molecules reflecting the free radical-induced arachidonic acid peroxidation, which can induce brain arteriole vasoconstriction [24,25]. Clinical observations seem to confirm an unprofitable effect of high blood oxygen tension, suggesting hyperoxia may be an independent risk factor for postoperative delirium in cardiac surgery patients [19,24]. Hence, it can be speculated that both hypoxia and hyperoxia may be associated with increased risk of delirium.

## 3. Neuroinflammatory Hypothesis

Every systemic inflammatory event triggers the release of several pro- and anti-inflammatory mediators, which may affect neuronal activity. Experimental studies have documented that peripheral cytokines released following the systemic inflammatory response can penetrate the blood–brain barrier (BBB) directly via active transport or indirectly via vagal nerve stimulation, and this effect can be intensified by hypoxia [26,27,28,29]. Elevated plasma pro-inflammatory cytokine concentrations, such as IL-1β and tumor necrosis factor α (TNFα), activate receptors in the endothelial cells, which causes cyclo-oxygenase activation, resulting in increased BBB permeability [30,31]. Additionally, elevated plasma interferon-γ concentration following a general inflammatory response damages occludin (a tight junction protein), which enables macrophage transition to the intracellular space in the brain, and stimulates astrogliosis and microglial activation [31,32]. Of note, peripheral administration of lipopolysaccharide induces a rapid elevation of TNFα in the brain per se [33]. An increase in BBB permeability is associated with cerebral edema and activation of microglia, which play a crucial role in synaptic plasticity and produces behavioral adaptation to environmental signals. Microglial cells are the main macrophage cells representing the brain immune system. Activated microglia secrete proinflammatory cytokines, eicosanoids and excitatory amino acids and stimulate production of reactive oxygen radicals and nitric oxide. Microglial activation is also responsible for regenerative processes and releasing neuroprotective factors. Disorders in microglial signaling impair memory [34]. It has been suggested that sepsis-related cognitive decline results from neuroinflammatory cascade following microglial activation [35]. Similarly, microglial activation with BBB dysfunction following the systemic inflammatory response was observed in mice that underwent elective orthopedic surgery [36]. Activated microglia and inflammatory mediators released by them modulate cholinergic, β-adrenergic and GABA-ergic neurotransmission, as well as secretion of vasopressin, corticotropin-releasing factor and adrenocorticotropic hormone, leading in turn to non-traumatic neuronal injury in the brain [37]. Significant reduction in the risk of postoperative delirium in patients treated with anti-inflammatory medications seems to confirm this hypothesis [38,39]. Interestingly, statins also reduce the neuroinflammatory response following systemic inflammation and/or ischemia-related brain dysfunction. Statins have been suggested in the treatment of delirium [40]. Several clinical studies have confirmed a relationship between systemic inflammation with increased cytokines, particularly IL-6, which are associated with delirium in patients treated for hip fracture [41,42]. Therefore, the important role of neuroinflammation in the development of postoperative neurocognitive dysfunction has become apparent.

## 4. Neurotransmitter Disorders

The occurrence of delirium can also result from dysfunction of multiple neurotransmitter systems. Disorders of the cholinergic system have been suggested as a crucial pathomechanism for delirium. Activation of acetylcholine receptors is associated with better learning and memory, and an inhibition of postsynaptic acetylcholine muscarinic-1 receptor corresponds to cognitive dysfunction and hallucinations [43,44]. Indeed, post-synaptic muscarinic-1 receptors are responsible for perception, attention and cognitive function [44]. Cholinergic hypofunction in the basal forebrain results in vulnerability to the cognitive deficits and memory dysfunction following systemic inflammation [45]. Additionally, centrally administrated interleukin 1β impairs memory in a cholinesterase-sensitive manner with reduction of acetylcholine outflow [46]. On the other hand, the release of acetylcholine can decrease the neuroinflammatory response to systemic inflammation via inhibition of IL-6, IL-8 and TNF release [42,43]. Clinical study has shown a correlation between low blood acetylcholinesterase concentration and delirium in cardiac surgery patients, whereas others have negated such a relationship [47,48].

Risk for delirium is also related to an age-dependent loss of dopamine receptors, and the imbalance between dopamine synthesis and dopamine receptors leading to neuropsychological disorders. The dopamine (DA) receptors influence the activity of ion pumps affecting neuronal excitability in the brain. The DA-1 and the DA-2 receptors modulate intracellular calcium levels and their activation increases intracellular calcium in a different manner. The DA receptors affect behavioral and locomotion functions [48]. The activation of D-1 receptors produces maximal locomotor stimulation, whereas activation of D-2 receptors decreases dopamine release reducing activity [49]. The DA-3 receptors, which are mainly localized postsynaptically in the nucleus accumbens, inhibit locomotor function [50]. Thus, elevated cerebral dopamine may cause neurobehavioral changes with raised cognitive impairment, anxiety and working memory dysfunction in elderly patients [51,52]. Dopamine receptors play a crucial role in cortical acetylcholine release and systemic administration of dopamine 2 antagonists significantly attenuated acetylcholine efflux [53]. A clinical study has shown that dopamine infusion increased the risk for delirium in cardiac surgery patients in a dose-dependent manner [54].

The DA-2 receptors inhibit neuronal signals via regulation of gamma-amino butyric acid (GABA) release. Indeed, downregulation in GABA receptor sensitivity is also suggested as an important pathomechanism of delirium, particularly in patients with alcohol dependency [55]. GABA is the most important inhibitory neurotransmitter in the cortex, hippocampus, amygdala, basal ganglia, cerebellum, medulla and spinal cord [56]. GABA concentrations in the cerebrospinal fluid and plasma have been considered a useful marker of brain activity and delirium [57]. Moreover, GABA plays a crucial role in sleep regulation, and disorders in GABA activity following neuroinflammation may result in sleep deprivation, which is one of the most important risk factors for delirium [58].

Disorders in the glutamatergic system in the limbic area, which strongly contribute to depression and mood disorders, are another reason for delirium. The proinflammatory cytokines, which are released by activated microglia, reduce glutamate uptake via inhibition of glutamate transporters on glial cells leading in turn to an increase in extrasynaptic glutamate concentration [59]. This glutamate binds to N-methyl-D-aspartate (NMDA) receptors reducing synaptic neuroplasticity and neuronal activity via suppression of synthesis and release of brain-derived neutrophic factor [60]. This hypothesis seems to be confirmed by Wyrobek and colleagues’ clinical observation, who noted a relationship between the decline in plasma brain-derived neutrophic factor (BDNF) concentration and episodes of delirium in elderly patients >70 years old who underwent lumbar spine surgery [61]. The elevated glutamate concentration bound to extrasynaptic NMDA receptors also suppresses the mammalian target of rapamycin signaling pathway, which reduces the synaptic plasticity and consequently impairs memory and learning [62].

The hypothesis that serotonin (5-hydroxytryptamine (5-HT)) plays an important role in the development of delirium has been examined in several studies [63,64,65]. 5-HT is produced from tryptophan by hydroxylation followed by acetylation and methylation to melatonin in the pineal gland. This last step is vitamin B6 dependent. Currently, 5-HT is widely distributed in the brain, and seven types of serotonin receptors have been characterized [64,65]. The serotonin 1 and 3 receptors (5-HT_1_ and 5-HT_3_) are responsible for learning and memory, the 5-HT_2_ receptors are responsible for cognitive function, the 5-HT_4_ receptors are responsible for disorders in the mood and depression development, and 5HT_7_ receptors are responsible for circadian rhythm [65]. The inhibition of serotonergic neurotransmission intensifies impulsivity and reduces patience [63,65]. The decrease in 5-HT synthesis specifically impairs short-term and long-term memory [63,64]. Reducing serotonin availability in the brain leads to delirium-like syndromes [66]. Clinical study with positron-emission tomography showed an age-dependent decrease of 1% per decade in striatal 5-HT_4_ receptors, and 13% lower 5-HT_4_ receptor activity in the limbic system with the largest difference of 19% in the amygdala in women compared with men [67]. Given the prevalent influence of 5-HT on emotion, memory, learning and circadian rhythm, the hypothesis about their crucial role in the pathogenesis of delirium seems to be correct. Interestingly, stress and inflammatory states are triggers for serotonin deficiency, and disorders in the above described receptors are initiated by neuroinflammation following trauma or general inflammatory responses [68,69].

## 5. Tryptophan Metabolism and Kynurenine Pathway Dysregulation

Tryptophan is an essential amino acid, which is metabolized within two main pathways: the serotonin pathway and the kynurenine pathway, leading to the synthesis of several neuroactive metabolites such as kynurenic acid (KYNA), 3-hydroxyanthranilic acid oxygenated to quinolinic acid (QUIN), picolinic acid, 5-hydroxyanthranilic acid, xanthurenic acid (XAN), kynurenine (KYN) and others (Figure 1). Some of the final metabolites in the kynurenine pathway present anti-excitatory activity, whereas others present pro-excitatory and pro-convulsive properties [68,70]. KYNA, a broad-spectrum antagonist of endogenous excitatory amino acids with generally accepted neuroprotective activity, blocks the strychnine-insensitive glycine recognition site in the NMDA receptor and the choline-induced increase in GABAergic function at the nanomolar and micromolar, physiological concentrations, respectively [71,72]. Elevated KYNA concentration was associated with myelin damage leading to neuronal dysfunction [73]. Indeed, accumulation of brain KYNA concentration induces learning and memory function, and a reduction of its level significantly improves cognitive function [74,75]. Therefore, it may be speculated that the physiological KYNA concentration in the brain has pronounced neuroprotective properties, whereas its elevated level induces cognitive disorders.

Another kynurenine pathway’s metabolite, QUIN, has neurotoxic activity via an increase in glutamate activity in the synaptic space by reducing the reuptake of glutamate in the presynaptic NMDA receptors. QUIN is co-localized with hyperphosphorylated tau protein and induces its phosphorylation in the cortical neurons [76]. A clinical study has documented a strong correlation between cerebrospinal QUIN concentration and the presence of dementia in AIDS patients [77]. Low plasma KYNA concentration and a marked increase in QUIN concentration is associated with a high risk for severe dementia in Alzheimer’s disease [78]. Several studies have documented that disorders of the tryptophan pathway are associated with memory dysfunction, dementia and delirium [79,80]. Additionally, QUIN is produced by activated macrophages, whereas elevated levels of KYNA may result from an inflammatory response [77,81]. Indeed, under stressful and inflammatory conditions tryptophan is quickly metabolised by indoleamine 2,3-dioxygenase (IDO), which plays a crucial role in the kynurenine pathway, and which is localized in the lung, the brain, the kidney and immune cells [82].

IDO activation has been shown to down-regulate neuroinflammation [83]. QUIN, picolinic acid, 3-hydroxykynurenine and 3-hydroxyanthranilic acids are neurotoxic and can cross the BBB during systemic inflammation [83,84]. Concededly, KYNA has no ability to cross the healthy BBB, however, general inflammation and inflammatory-related endothelial activation lead to BBB injury, opening the way for penetration of high amounts of KYNA to the brain [85]. KYNA is also produced and released by inflammation-activated astrocytes and microglia [86]. Therefore, we suggest that intra-cerebral activation of the kynurenine pathway and cerebral influx of neurotoxic kynurenine metabolites may lead to neuronal damage and in turn delirium. However, these pathomechanisms should be confirmed in further studies.

## 6. Gut Microbiota Dysregulation

Recently, the role of gut microbiota as a physiological regulator of essential processes including brain function has been the subject of many investigations. It has been documented that the brain–gut axis, a complex bi-directional signaling system, regulates brain function [87]. Abnormal composition of intestinal microbiota may contribute to the development of neurodegeneration and neuroinflammation, which are associated with depression or autism [88,89]. An experimental study has presented a modification of gut microbiota following gastrointestinal surgery, which required long-time transformation [90,91,92]. Changes in gut microbiota composition mainly included *Enterobacteriaceae*, *Bacteroidaceae* and *Rhodospirillaceae* [91]. This microbial dysbiosis is closely linked to disturbances of gene expression of inflammatory cytokines. It has been documented that abnormal intestinal microbiota composition may be an important risk factor of postoperative delirium [93]. An experimental study has shown an elevated amount of *Escherichia* and *Shigella* in animals with delirium [94]. A clinical observation seems to confirm this relationship, because mechanically ventilated patients with delirium had increased amounts of *Firmicutes* bacteria with concurrent reduction of *Proteobacteria* in the gut, and these changes did not correspond to early nutrition, microbiome composition, and the type of delirium [95]. Interestingly, treatment of postoperative dysbiosis with *Lactobacillus* or other probiotics mitigated delirium [95]. It has been documented, that disorders in gut microbiota are associated with the severity of depressive syndromes [96,97]. Hence, it can be suggested that the microbiome affects neurocognitive dysfunction, however this hypothesis needs further study.

## 7. Treatment of Delirium

Independent of the type of delirium, early identification of the risk factors of non-traumatic brain injury and their modification or elimination are the most important elements of management for reduction of delirium severity. It has been documented that approximately 30% of delirium cases are preventable [98]. Prevention and treatment of delirium should be based on the implementation of the routine daily practice care bundle presented by the Society of Critical Care Medicine (SCCM) called the ABCDEF bundle (A—Assess, Prevent, and Manage Pain, B—Both Spontaneous Awakening Trials (SAT) and Spontaneous Breathing Trials (SBT), C—Choice of analgesia and sedation, D—Delirium: Assess, Prevent, and Manage, E—Early mobility and Exercise, F—Family engagement and empowerment) [99]. The implementation of early mobility activities combined with an appropriate level of sedation and adequate pain management is a challenge in critically ill patients, but the combined efforts seem to be effective methods of delirium prevention and treatment [100,101]. It must also be emphasized that the mainstay of delirium treatment is early and focused management of disrupted homeostasis that can lead to delirium. This approach should focus on treatable or reversible conditions, including treatment of hypoxia, correction of underlying electrolyte disorders (i.e., hypo- or hypernatremia), early detection and treatment of infections, maintaining adequate volemia and preventing gastrointestinal disorders.

Adequate management of hypoxia in a patient with delirium should be the primary goal of the ICU team. It should not only be based on providing supplemental oxygen or mechanical ventilation, but also on ensuring adequate cerebral blood flow, avoidance of anemia or avoidance of a range of factors potentially leading to cerebral vasoconstriction (i.e., hypocarbia). Continuous monitoring of cerebral oxygenation seems to reduce the risk of delirium effectively. Different clinical studies and meta-analyses have shown close relationships between disorders in cerebral oximetry and the severity of delirium, suggesting that cerebral oximetry is an easy and feasible method to measure risk of postoperative neuropsychological disorders in cardiac surgery patients [24,102,103,104]. Likewise, continuous measurement of cerebral oximetry can help to identify an episode of cerebral hypoxia allowing quick intervention, which may reduce the risk of delirium in critically ill ICU patients [16,105]. Hence, monitoring of cerebral oximetry should be widely used in clinical practice, especially in patients at increased risk of delirium.

Monitoring of hyperoxemia is difficult and commonly requires regular blood gas analysis because pulsoximetry and arterial saturation (SpO_2_ and SaO_2_, respectively) are not credible when arterial partial oxygen pressure (PaO_2_) increases above 100 mmHg. Recently, the oxygen reserve index (ORI) has been implemented into clinical practice to avoid hyperoxemia [106,107]. The ORI is a new multiple-wavelength pulse oximetry reflecting the oxygenation status in the moderate range of hyperoxia with a PaO_2_ of approximately 100–200 mmHg [106]. Although this technology is new and not commonly applied to routine clinical practice, it seems to make oxygen therapy significantly safer and easier. Nevertheless, the usefulness of the ORI in prevention of hyperoxia-related delirium requires further studies.

Maintaining the circadian rhythm of wakefulness and sleep in the ICU is difficult; therefore, attention to normalize the circadian rhythm is of uttermost importance. It has been shown by Skrobik et al. that introducing a low nocturnal dose of dexmedetomidine reduces the incidence of delirium but does not improve sleep quality [108]. Treatment with melatonin to correct the circadian rhythm is commonly used in patients with elevated risk of delirium [109,110,111]. Physiologically, melatonin secretion is low during daytime and increases early in the evening and at night with the peak in the middle of the night [112]. The circadian rhythm of melatonin secretion inversely corresponds to cortisol secretion [113]. The desynchronization of the melatonin secretion rhythm has been reported in sedated critically ill patients [110]. This desynchronization may result from disturbances in cortisol secretion in critically ill patients or may be associated with a disturbed production of tryptophan, because an elevated level of plasma interferon-γ following an inflammatory response induces IDO activity leading to intensive tryptophan degradation in the kynurenine pathway [114,115]. Notably, increased IDO activity has been described in patients with major depression [115,116]. Hence, treatment with melatonin should be implemented in depressive critically ill patients, in whom hypoactive or mixed delirium has been diagnosed.

Immuno-inflammatory activation also plays a crucial role in the pathophysiology of major depression, and elevated levels of inflammatory markers have been noted in patients with delirium [117,118,119]. Therefore, some advocate the use of anti-inflammatory medications for consideration in patients with non-traumatic brain injury. Importantly, treatment with dexamethasone did not reduce the incidence of delirium, and intra-articular administration of corticosteroids induced hyperactive delirium in elderly patients with moderate dementia [120,121]. Experimental and clinical studies documented the anti-inflammatory and immunomodulatory effect of statins, which was associated with a reduction of delirium in critically ill patients [40,122,123,124,125]. Administration of atorvastatin/simvastatin decreased systemic and brain tissue levels of proinflammatory cytokines and reduced lipid peroxidation, preventing the development of long-term cognitive dysfunction [122]. Another study also documented that simvastatin reduced the severity of depression via reduction of microglia and astrocyte activation in the hippocampus after experimental traumatic brain injury [124]. Based on these observations, some authors postulate a protective effect of statins connected with their anti-inflammatory activity in the brain. The possibility of a therapeutic anti-neuroinflammatory effect of statins in patients with delirium should be confirmed in future clinical trials.

Treating agitation in the ICU has always been challenging and difficult, therefore antipsychotics and anti-convulsive medications are commonly used in patients with hyperactive and mixed delirium [126,127,128,129]. Antipsychotics are thought to work by nonspecific blockade and restoration of the imbalanced neurotransmission in the brain. Haloperidol, the most popular neuroleptic agent, is not recommended for routine use in delirium [130] but may have a role in the hyperactive subtype. If used it should only be continued until agitation is controlled and no longer thereafter, because it may cause several extrapyramidal symptoms and may be associated with increased mortality in elderly hospitalized patients [126]. A clinical study including 68 mechanically ventilated patients with subsyndromal delirium documented that a low dose of haloperidol administrated early during the ICU stay did not prevent delirium and had little therapeutic advantage [127]. A retrospective analysis of the effectiveness in the treatment of delirium showed no significant differences in delirium duration and secondary outcome in geriatric patients treated with different antipsychotic agents [128]. A comparison of haloperidol, ziprasidone and placebo in a randomized, double-blind trial of 566 patients with ICU delirium found no effect of either antipsychotic medication on the number of coma-free and delirium-free days [131]. Of note, antipsychotics also present extracerebral adverse effects such as QTc prolongation, which is an independent risk factor for life threatening cardiac arrhythmia and sudden cardiac death [132]. Antipsychotics cannot be suggested as the first line in treatment of delirium. Valproic acid presents similar effectiveness to antipsychotics in the treatment of agitation associated with hyperactive delirium, and the adverse effects of its administration are lower than antipsychotics; overall this agent is well tolerated [129,133]. Hence, the use of valproic acid as an adjuvant for treatment of hyperactive delirium can be a promising alternative to antipsychotics.

It must be stressed that effective delirium management should be based on its prevention and the use of non-pharmacological measures, rather than pharmacological treatment. Management strategies should include noise reduction, exposure to natural light during the day, limiting exposure to light at night, avoiding extremes of temperature, ensuring undisturbed night rest. It is extremely important to ensure efficient communication with the environment, which includes the patient’s daily orientation in time, place and the condition and support of their senses (e.g., provision of glasses and hearing aids). Moreover, the presence of family and friends to support the patient at the bedside and provide a link with the reality outside of the ICU is of uttermost importance when dealing with delirium in critically ill patients.

## 8. Conclusions

This review briefly presents the most important pathomechanisms for delirium in critically ill patients. Many of them are the basis for the application of targeted treatment in delirium. However, the effect of pharmacological treatment may depend on brain reserve, cognitive reserve, intellectual and non-intellectual activity and severity of pre-existing drug or alcohol addiction (if present) [134]. Therefore, the detailed recognition of the pathomechanisms of the non-traumatic brain injury requires further study and shall lead to effective therapeutic options.

## Figures and Tables

**Figure 1 medicina-56-00469-f001:**
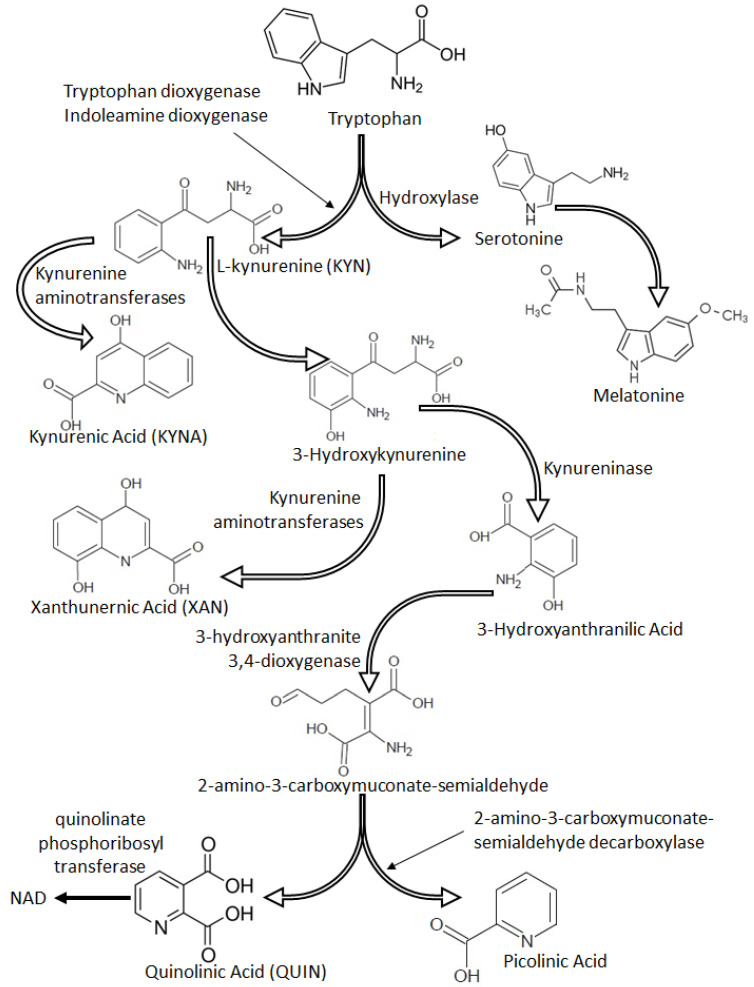
Tryptophan pathways.

**Table 1 medicina-56-00469-t001:** Studies regarding pathomechanisms of delirium—design and main characteristics.

Pathomechanisms	Authors and Reference Number	Study Design	Number of Patients	Results
Hypoxia	Funk et al. [16]	Prospective controlled clinical study	15 septic shock patients	Decrease in cerebral saturation corresponds to the incidence of delirium
	Mikkelsen et al. [17]	Prospective, multicentre cohort clinical study	406 adult patients treated for ARDS	Low PaO_2_ was associated with cognitive impairment
	Hopkins et al. [18]	Prospective controlled clinical study	120 adult patients treated for ARDS	Hypoxia assessed as SaO_2_ < 90% is associated with long-term neurocognitive disorders
Hyperoxia	Kupiec et al. [19]	Retrospective clinical study	93 cardiac surgery patients	Hyperoxia defined as PaO_2_ > 120 mmHg is associated with the occurrence of postoperative delirium
	Mutch et al. [20]	Prospective clinical study	12 healthy volunteers	Disturbance in cerebral blood flow following hyperoxia corresponds with postoperative neuropsychological disorders
	Lopez et al. [24]	Prospective controlled clinical study	310 cardiac surgery patients	Hyperoxia defined as any intraoperative cerebral oxygenation greater than baseline
Neuroinflammation	Velagapudi et al. [36]	Experimental, behavioural and histological study	61 animals undergoing orthopaedic surgery	Orthopaedic surgery leads to microglial activation, astrogliosis and brain blood-barrier disruption
Disorders in neurotransmitters	Adam et al. [47]	Prospective observational study	114 cardiac surgery patients	Decrease in acetylcholine hydrolysing enzyme activity increases risk for delirium
	John et al. [48]	Prospective observational study	251 cardiac surgery patients	There are no correlations between acetylcholine hydrolysing enzyme activity and risk of delirium
	Yilmaz et al. [54]	Prospective observational study	137 cardiac surgery patients	Dopamine infusion is an independent risk factor for delirium
	Yoshitaka et al. [57]	Prospective observational study	40 critically ill patients	Plasma GABA activity is associated with delirium
	Wyrobek et al. [61]	Prospective observational study	77 elderly patients undergoing spinal surgery	Decrease in the brain-derived neurotrophic factor is associated with delirium
	Madsen et al. [67]	Prospective observational study	30 healthy volunteers	Disorders in 5-HT4 receptor correlate with impaired memory and risk for neuropsychiatric disorders
Tryptophan metabolism and kynurenine pathway dysregulation	Kozak et al. [75]	Experimental, behavioural and histological study	Animal study	Elevated brain kynurenic acid impairs cognitive function
	Valle et al. [77]	Prospective observational study	62 HIV-infected patients	Elevated quinolinic acid is a risk factor for neurocognitive disorders
	Gulaj et al. [78]	Prospective observational study	34 patients with Alzheimer dementia	Plasma kynurenic acid and quinolinic acid correlate with impaired cognitive function
	Solvang et al. [79]	Prospective observational study	155 patients with dementia	Kynurenine had a nonlinear quadratic relationship with cognitive disorders
Gut microbiota dysregulation	Zhang et al. [93]	Experimental, behavioural study	11 pigs	Gut microbiota disorders induce delirium
	Liufu et al. [94]	Experimental, behavioural study	10 mice	Gut microbiota disorders induce delirium
	Liskiewicz et al. [96]	Prospective observational study	16 patients with major depression	Disorders in gut microbiota are associated with the severity of depression
	Huang et al. [97]	Prospective observational study	54 patients with major depression	Defects of the Firmicutes (gut bacteria) increase a risk for depression

Legend: ARDS—adult respiratory distress syndrome; GABA—gamma-amino butyric acid; HIV—human immunodeficiency virus; 5-HT4—5-hydroxytryptamine; PaO_2_—partial pressure of oxygen; SaO_2_—oxygen saturation.

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
