# Peer review of "Pathomechanisms of Non-Traumatic Acute Brain Injury in Critically Ill Patients"

_medicina, 2020, doi:10.3390/medicina56090469_

Round 1
Reviewer 1 Report
I read the manuscript “Pathomechanisms of non-traumatic acute brain injury in critically ill patients” from Dabrowski W. et al. with great interest.
By this well-written review article the authors give an excellent overview about relevant pathomechanisms related to the highly important field of delirium in critically ill patients.
I have no concerns.
Author Response
Thanks
Reviewer 2 Report
Dear authors
This is a good work
Author Response
thanks
Reviewer 3 Report
More attention is needed to spelling and English grammar (eg ref sp Psychiatry etc)
A table of each mechanism contributing to delirium will help clarify. If so which Rx is specific
The final section on 'natural medicines' is not evidence based. It is overly speculative: would not pass muster for therapy as recommendations doses etc are highly non specific. It would be better to omit this section all together and focus on conventional therapy for each aspect of delirium: mainly oxygen and correction of metabolic disorders
Author Response
Dear Reviewer.
We thank the reviewers for their review of our manuscript. We have corrected our manuscript according to the reviewers' comments. We deleted the last paragraph, added the table with each mechanism contributing to delirium, and added figure 1 with the tryptophan pathway. We've submitted the corrected version of our manuscript.
Sincerely Yours
Wojciech Dabrowski on behalf of all co-authors
Round 2
Reviewer 3 Report
Now that the authors have eliminated the section on natural medicines, it would be advisable to discuss treatment of the three different subtypes based upon their elaborate phenomenology. Exactly how does this analysis advance diagnosis and treatment?
Author Response
Dear Reviewers
We thank the reviewers for their helpful follow-on second review of our manuscript medicina-901196 under the title: "Pathomechanisms of non-traumatic acute brain injury in critically ill patients". We adjusted the manuscript per the reviewers' additional comments. We added a short section with treatment of delirium, in which we generally described a treatment from the pathomechanisms as well as the type of delirium. We hope that our manuscript meets the reviewers’ requirements
Wojciech Dabrowski on behalf all co-authors
Round 3
Reviewer 3 Report
The paper still contains many misspellings eg stanins for statins and others. I suggest the authors go back and take time to assure accuracy and not to impose a hastiness to publish upon a reading public. The treatment section begins to improve but the most notable treatments oxygen for hypoxia and correction of underlying electrolyte disorders, needs emphasis over eg melatonin. I do not believe correcting sleep disorders is "easy". May I request that the authors take more time and respond in a thoughtful, precise and considered manner with a revision suitable for consideration as an addition to the medical scientific literature. This MS needs more time and work.
Author Response
Dear Reviewer
Thank You for Your usable comments. We corrected our manuscript adding some sentences with general rules of delirium treatment and blood oxygen monitoring. We hope, our amendments improve the paragraph describing delirium treatment. We also correct the English language.
Sincerely Yours
Wojciech Dabrowski on behalf of all co-authors